# Coupled Biospheric Synchrony of the Coastal Temperate Ecosystem in Northern Patagonia: A Remote Sensing Analysis

**Carlos Lara [1,2]\* [ID], Bernard Cazelles [3,4,5] [ID], Gonzalo S. Saldías [2,6] [ID], Raúl P. Flores [7] [ID], Álvaro L. Paredes [8] [ID] and Bernardo R. Broitman [9] [ID]**

[1]   Centro de Investigación en Recursos Naturales y Sustentabilidad, Universidad Bernardo O'Higgins, 8370993 Santiago, Chile

[2]   Departamento de Física, Facultad de Ciencias, Universidad del Bío-Bío, Casilla–5C, 4051381 Concepción, Chile; gsaldias@ubiobio.cl

[3]   iGLOBE, UMI CNRS 3157, University of Arizona, Tucson, AZ 85719, USA; cazelles@biologie.ens.fr

[4]   UMMISCO, UMI 209, Sorbonne Université-IRD, 75006 Paris, France

[5]   IBENS, UMR CNRS 8197, Eco-Evolution Mathématique, Ecole Normale Supérieure, 75005 Paris, France

[6]   Centro FONDAP de Investigación en Dinámica de Ecosistemas Marinos de Altas Latitudes (IDEAL), 5090000 Valdivia, Chile

[7]   Departamento de Obras Civiles, Universidad Técnica Federico Santa María, 2390123 Valparaíso, Chile; raul.flores@usm.cl

[8]   Programa de Magíster en Estadística, Facultad de Matemáticas, Pontificia Universidad Católica de Chile, 8331150 Santiago, Chile; aaparede@mat.uc.cl

[9]   Departamento de Ciencias, Facultad de Artes Liberales, Universidad Adolfo Ibáñez, 2562340 Viña del Mar, Chile; bernardo.broitman@uai.cl

\*   Correspondence: carlos.lara@ubo.cl

**Abstract:** Over the last century, climate change has impacted the physiology, distribution, and phenology of marine and terrestrial primary producers worldwide. The study of these fluctuations has been hindered due to the complex response of plants to environmental forcing over large spatial and temporal scales. To bridge this gap, we investigated the synchrony in seasonal phenological activity between marine and terrestrial primary producers to environmental and climatic variability across northern Patagonia. We disentangled the effects on the biological activity of local processes using advanced time-frequency analysis and partial wavelet coherence on 15 years (2003–2017) of data from MODIS (Moderate Resolution Imaging Spectroradiometer) onboard the Terra and Aqua satellites and global climatic variability using large-scale climate indices. Our results show that periodic variations in both coastal ocean and land productivity are associated with sea surface temperature forcing over seasonal scales and with climatic forcing over multi-annual (2–4 years) modes. These complex relationships indicate that large-scale climatic processes primarily modulate the synchronous phenological seasonal activity across northern Patagonia, which makes these unique ecosystems highly exposed to future climatic change.

**Keywords:** MODIS imagery; phenological cycle; Patagonia; climatic variability; wavelet analysis

---

## 1. Introduction

Over the last century, climate change has altered the physiology, distribution and phenology of marine and terrestrial species across the planet [1,2]. The imprint of climatic variability on phenological patterns such as flowering, fruiting and migration rates changes across functional groups and trophic levels [3–5]. However, the effects of climatic forcing on the variability of biological processes cascade

through a complex network of relationships that arises from primary productivity and has enormous adaptive and evolutionary importance in ecological systems [2].

The seasonal variation of environmental drivers imposes its rhythms and thus phenological synchrony, where spatio-temporal fluctuations of different ecological patterns are locked in phase across different ecosystems [6–8]. A substantial body of literature documents that different populations are influenced by coherent environmental fluctuations (e.g., [2,3,9–14]). For example, evaluating the effects of climate variation on the intraguild-predation structure of the fish population in the Windermere lake, Edeline et al. [12] show that the pathogen-induced regime shift is temperature-controlled. Regional changes in zooplankton biomass and the duration of upwelling events vary coherently along the Northwest Iberian shelf [13], while on a global scale, spatial synchrony in chlorophyll concentration has been reported for the world's oceans [14].

Similarly, synchrony between ecological processes and environmental forcing has been observed between coastal communities separated by a few hundred km [10]. While most of the literature has focused on the spatial scales of synchrony, i.e., the Moran effect [15], limited attention has been given to changes in synchrony over time either in marine [16,17] or terrestrial ecosystems [18]. Marine and terrestrial ecosystems in coastal areas are strongly coupled by fluxes of matter and energy. As a consequence, these areas are among the most productive regions on the planet, including environments such as coastal forests along temperate upwelling regions [19,20]. The elevated primary productivity in these narrow regions depends on the synchrony of marine and terrestrial processes resulting from a combination of environmental forcing, the biological responses to it and the linkages that result from fluxes between ecosystems [21]. As a result, the study of coupled temporal fluctuations between terrestrial and marine autotrophic biomass may represent a bellwether of the effects of climatic forcing on tightly coupled physical-biological processes shaped through evolutionary and adaptive processes.

The northern Patagonia ecosystem is a temperate coastal ecosystem on the edge of the southeast Pacific, where the storm track of the West Wind Drift Current impinges the shores of South America along the Valdivia Rainforest ecoregion [22]. The marine and terrestrial environments are influenced by large-scale climatic variability through changes in precipitation [23] and the properties of the water masses that flow alongshore [24,25]. The region sustains great food and forestry industries that are among the top five producers globally [26]. The synergistic effects of intense land use change and coastal aquaculture activities have degraded the region's ecosystems and reduced their resilience to external shocks [27,28].

Disruptions of critical ecosystem services have been recently reported, such as sharp drops in the availability of competent mussel spat that are seasonally collected by the aquaculture and used as seed stock. The limitation of larval supply appeared to be driven by a change in the phenology of surface chlorophyll–a concentration, which was linked to large-scale climate variability as El Niño-Southern Oscillation, Southern Annular Mode and Pacific Decadal Oscillation (ENSO, SAM and PDO, respectively) [16]. Similarly, climatic events seem to have triggered anomalous blooms of harmful algae [29], which took place near-simultaneously with events observed along the northwest Pacific [30]. Human intervention on land has led to changes in land use and land abandonment, which has altered soil fertility and water availability [31,32]. Land degradation may have also caused an increase in wildfire risk, impacting ecosystem services such as recreation and ecotourism [32]. Human alteration of coastal terrestrial ecosystems along the region has already translated into changes in the productivity of the marine environment [33]. However, the understanding coupled biological changes at different spatio-temporal scales is limited by the lack of ecological data and integrated studies of these ecosystems, which are also threatened by local anthropogenic activities and exposed to large-scale climatic variability (e.g., ENSO, SAM).

The use of satellite–derived data (e.g., Moderate Resolution Imaging Spectroradiometer– MODIS) has become a powerful tool to monitor ecological dynamics such as vegetation biomass or surface chlorophyll–a distribution at regional and global scales at a very low cost [34]. In this study, we take advantage of the capabilities of satellite-derived data and explore the spatial and temporal co–variation

in phenological patterns across the marine and terrestrial ecosystems in northern Patagonia (41–44°S). We aim to improve our understanding of the effects of environmental forcing on patterns of productivity along these productive coastal zones. We consider the time series from the Enhanced Vegetation Index (EVI), surface chlorophyll–a concentration (Chl–a), and the normalized fluorescence line height (nFLH) in order to test a simple hypothesis: the synchrony of seasonal biological activity in the northern Patagonia ecosystem is related to climate variability over the southern Pacific Ocean. A description of the satellite data and the statistical analyses used in the paper are presented in Section 2. The main results are described in Section 3, and a discussion of the principal modes of synchrony and the evolution of correlation between temporal series across multiples scales is presented in Section 4. Finally, the conclusions are presented in Section 5.

## 2. Methodology

### 2.1. Study area

The Northern Patagonia ecosystem is located in southern Chile (Figure 1A), including the Chiloé island and a group of 40 smaller islands (Figure 1B). Native forests cover 66.9% of the total land area, while 27.4 % corresponds to shrubland and agricultural land [35]. A large part of the southern sector of Chiloé island (46.5%, see Figure 1) is protected natural areas, which are distributed among several National Parks and Ecological Reserves. The primary forest type is broad-leaved evergreen. Although the territory is sparsely populated, the forest stands of northern Chiloé forests have been severely exploited to support domestic fowl, grazing and farming [36]. Aquaculture is the most important economic activity in this region with a total mussel production of 283,307 $t/y^{-1}$ in 2014 [37].

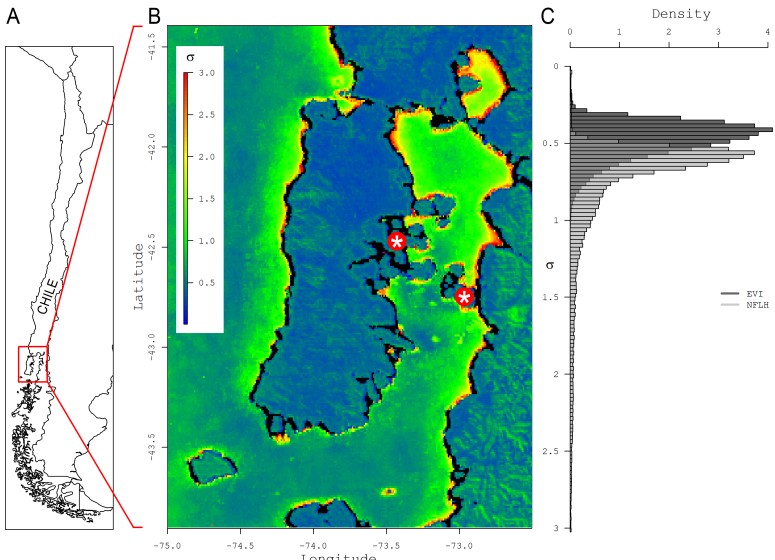

**Figure 1.** Study area: (**A**) Study area showing its location along the coast of Chile. (**B**) northern–Patagonia Ecosystem coloured using the standard deviation of Enhanced Vegetation Index (land) and normalized fluorescence line height (water) from all available images over the 2003–2017 study period. (**C**) Histogram of the standard deviation in EVI (black) and nFLH (gray). The full temporal series of images (2003 to 2017, for EVI and nFLH) was used to calculate one single image that contained the variability of both measurements. First, every image was standardized to work with both sets in the same units, using the formula $\frac{x - mean(X)}{sd(X)}$. For EVI, the image was re-sampled to 1km, to match the resolution of nFLH. Then, for every stack of images (EVI and nFLH), the standard deviation of every pixel trough time is calculated and a single image is generated for every product. Both images, the standard deviation for EVI and nFLH, are added in order to fill the land/water that both products do not cover. In all points where they overlap (coastal zones) the values were filled with null values. The main axis of Desertores Islands, are denoted by red asteriks.

Maximum and minimum temperatures occur during the austral summer and winter, respectively, and the yearly precipitation exceeds 5000 mm [38]. The Inner Sea of Chiloé includes embayments, fjords, channels, straits and estuaries that are characterized by shallow bathymetry (50–250 m deep); they receive oceanic water from the Pacific Ocean and significant freshwater input from precipitation and river discharges. River discharge in this region follows a nival regime and comes primarily from the Petrohué, Cochamó, Puelo, Riñihue and Yelcho Rivers, located in our study region. The combined freshwater discharge from these rivers influences local ocean circulation and the spring–autumn phytoplankton bloom dynamics [25,39]. Due to the topographic and bathymetric complexity and the distinct environmental regimes in the northern Patagonia ecosystem [16], we separated the Chiloé Island and Inner Sea of Chiloé regions into two areas: north (41–42.5°S) and south (42.5–44°S).

## 2.2. Satellite Data

To characterize temporal variation in terrestrial vegetation, we used 15 years (2003–2017) of Enhanced Vegetation Index (EVI) derived from MODIS–Terra, with a temporal and spatial resolution of 16 days and 250 m (MOD13Q1 version 6) [40]. The EVI index was developed to optimize the detection of vegetation signals, improving sensitivity in high biomass regions, and reducing atmospheric influence [41]. The EVI index is based on 469, 645 and 858 MODIS wavelength bands and has shown better performance than the Normalized Difference Vegetation Index in regions with high vegetation density such as our study region [39,41].

To study the dynamics of marine phytoplankton biomass, we used Ocean Color (OC) data from MODIS-Aqua (e.g., [42]), which were processed using the SeaDAS (SeaWIFS Data Analysis System version 7.5.1) software. The images produced for the 2003–2017 period were used to generate 1 km resolution daily images, completing 15 years of high-resolution observations. The processing of the OC data was done using default atmospheric corrections [43]. Although the black pixel assumption does not hold when using NIR (Near InfraRed) bands for atmospheric corrections when the water is turbid, we chose to maintain the default (NIR) settings because of the implementation of SWIR (Short Wave InfraRed) bands [44] in SeaDas increased the noise. The OCx chlorophyll–a (in mg m$^{-3}$) is a fourth-degree polynomial algorithm based on empirical regression between blue/green $R_{r}s$ ratios. Following high levels of organic matter and the noise:error ratios associated with OC in Northern Patagonia ecosystem [16], we complemented our estimates of primary production in the marine environment using the normalized Fluorescence Line Height (nFLH). nLFH is calculated as the difference between the observed nLw(678) and a linearly interpolated nLw(678) from two surrounding bands [45]. To evaluate the local effect on phenological patterns, we also generated a monthly composite of daytime Sea Surface Temperature (SST) derived from MODIS-Aqua. The SST product is retrieved from MODIS bands 31–32 (10.8–12.3 µm) [46]. The histograms in Figure 1C show the frequency distribution of observations over land (EVI) and over water (nFLH) for the image shown in Figure 1B, to highlight the near-gaussian distribution of EVI and the positive skewness of nFLH. These results show that water has more extreme values of biological production (concentrated on coastal areas) compared to land, that tends to be smoother in this area, without big peaks on EVI variation.

## 2.3. Climate Indices

To understand the impact of climatic-scale processes on primary production we used the El Niño-Southern Oscillation (ENSO) and Southern Annular Mode (SAM) climate indices, which have been shown to influence the regional climate in the Southern Hemisphere [47,48]. Monthly values of SOI (Southern Oscillation Index) are based on the calculation by [49], whereas SAM monthly data are based on the methodology proposed by [50].

## 2.4. Wavelet Analysis

The satellite-derived time series for the 2003–2017 period for Chl-a, nFLH, EVI, and climatic indices were analyzed to assess their patterns of inter and intra-annual variability and to investigate

possible differences in their seasonal phenological patterns. Continuous wavelets were used to examine the temporal periodicity and their time evolution [51]. Wavelet analysis offers a robust method to decompose time series data into the time–frequency domain to explore the dominant modes of variance within a series [51]. Thus it is especially appropriate for the analysis of biosphere time series (e.g., Chl-a, nFLH, EVI, SST) due to the non-stationary character of ecological signals (e.g., phenological patterns). Transient dynamics can play a crucial role in the spatio-temporal structure of such variables, and wavelet analysis can illustrate shifts in the scale of variance through the time series [52].

In this study we used the Morlet wavelet

$$\psi(t) = \pi^{-1/4} exp(-i\omega_0 t) exp(-t^2/2) \tag{1}$$

The Morlet wavelet is defined as the product of a complex sinusoidal, $exp(-i\omega_0 t)$, and a Gaussian envelope, $exp(-t^2/2)$, with a central angular frequency of $\omega_0$ [52]. The relative importance of each frequency at each time is represented in the time-frequency domain, forming the local wavelet power spectrum (WPS). We also computed the global WPS as the time-average of the local WPS for each frequency component, which provides an unbiased spectrum that allows determining the significant temporal cycles of the time series [52].

The wavelet transform of temporal series $x(t)$ with respect to the mother wavelet is obtained as:

$$W_x(\alpha, \tau) = \frac{1}{\sqrt{\alpha}} \int_{-\infty}^{\infty} x(t)\psi^*(\frac{t-\tau}{\alpha})dt = \int_{-\infty}^{\infty} x(t)\psi_{\alpha,\tau}^*(t)dt \tag{2}$$

where $^*$ denote the complex conjugate form, $W_x(\alpha, \tau)$ is the contribution of the scale $\alpha$ to the signal at different position $\tau$ [51,52]. With continuous wavelets there is a direct mathematical relationship between scale $\alpha$ and the frequency $f$, thus one can replace $\alpha$ by the frequency $f$ (see [51]).

*2.5. Coherence and Synchrony Analysis*

The transient patterns of co-variation between the EVI and Chl-a were analyzed through wavelet coherence, which quantifies the association between signals as a function of frequency and time. The wavelet coherence is defined as the wavelet cross-spectrum normalized by the spectrum of each of the signals:

$$WC_{x,y}(f, \tau) = \frac{\|< W_{x,y}(f, \tau) >\|}{\|< W_x(f, \tau) >\|^{1/2}\|< W_y(f, \tau) >\|^{1/2}} \tag{3}$$

where $<>$ denotes a smoothing operator in both time and space, $W_{x,y}(f, \tau)$ is the wavelet co–spectrum of the two time series $x(t)$ and $y(t)$, and $W_x(f, \tau)$ and $W_y(f, \tau)$ are the wavelet transform of the $x(t)$ and $y(t)$, respectively. $WC_{y,x}(f, \tau)$ equals one when there is a perfect linear relationship at a specific time $t$ and frequency $f$ between the two signals, and equals zero if the time series are independent [9,52].

It is important to control for dependence between environmental signals when analyzing their influence on a given ecological system. To that end we used Partial Wavelet Coherence (PWC), which complements the wavelet power spectrum and wavelet coherence. PWC computes the coherence between two signals after having controlled the effect of other signals. For instance, Ng and Chan [53] used the iterative PWC method for computing the coherence between signals $y$ and $x_1$ controlling the common effect due to $x_2$:

$$PWC_{y,x_1,x_2}(f, \tau) = \left( \frac{|WC_{y,x_1}(f, \tau) - WC_{y,x_2}(f, \tau)WC_{y,x_1}^*(f, \tau)|^2}{|1 - WC_{y,x_2}(f, \tau)|^2|1 - WC_{x_2,x_1}(f, \tau)|^2} \right)^{1/2} \tag{4}$$

Nevertheless, this iterative method is not appropriate to analyze more than three–times series. Thus, we adapted the inverse PWC method for Fourier analysis [54] to wavelet analysis. This inverse

PWC method is based on the spectral matrix $\mathbf{S}(f)$ where all the elements are the cross-wavelet spectrum for the $i$ and $j$ signals from the $n$ time series analyzed $x_n(\mathrm{t})$:

$$\mathbf{S}(f) = \begin{pmatrix} W_{11}(f,t) & W_{12}(f,t) & \ldots & > W_{1n}(f,t) \\ W_{21}(f,t) & W_{21}(f,t) & \ldots & W_{2n}(f,t) \\ > \vdots & \vdots & \ddots & \vdots \\ W_{n1}(f,t) & W_{n2}(f,t) & > \ldots & W_{nn}(f,t) \end{pmatrix} \tag{5}$$

The PWC between $x_j(t)$ and $x_k(t)$ controlled by all other time series is then defined as:

$$PWC_{jk|(\backslash jk)}(f,t) = \left( \frac{|\, S^{jk}(f)\,|^2}{|\, S^{jj}(f)\,|^2 |\, S^{kk}(f)\,|^2} \right)^{1/2} \tag{6}$$

where $S^{jk}$ is the $(j,k)$ element of the inverse spectral matrix $\mathbf{S}^{-1}(f)$ and $(\backslash jk)$ means all elements excepted the jth and the kth.

Satellite data were separated into northern and southern regions based on latitude; pixels located between 41–42.5°S were assigned to the northern region, whereas pixels located between 42.5–44°S were assigned to the southern region. The time series were obtained by averaging, at each time step, all pixels corresponding either to the northern or southern regions. Thus, we obtained single time series that are representative of the entire northern and southern regions. Special care was placed on excluding pixels with no data in order to avoid biases on the average values used in the time series. All time-series data were aggregated into monthly intervals and standardized for the joint analyses.

## 3. Results

### 3.1. Temporal Variability in the Northern–Patagonia Ecosystem

The significant frequency modes for all variables are presented in Figure 2. The local WPS for Chl–a in the northern area (Chl-a–North) indicated significant semiannual (0.5-year) and inter-annual (3-year) periods (Figure 2A), which appeared in different parts of the record. A significant annual period was evident for Chl–a the southern area (Chl-a–South). The interannual component was intermittent and absent over the 2010–2013 interval (Figure 2B). Examination of the local WPS of nFLH in the northern area (nFLH–North) showed that a significant semiannual oscillation took place between 2006–2007, and also during 2016. The annual component was significant during 2006–2008 and 2012–2016 periods (Figure 2C), while different dominant periods were observed in the southern area (nFLH–South, Figure 2D); the local WPS showed a semiannual period during 2006–2007 and 2015–2017 intervals, and an annual periodic oscillation during the 2011 and 2015–2017 intervals. Besides, a significant interannual signal (2-year) was observed between 2005–2007. We note that the subdecadal and subannual modes need to be interpreted with caution due to the short duration and low temporal resolution of the time series used in the analysis, respectively. The temporal dynamics of EVI in both northern and southern areas (EVI–North and EVI–South)showed that the annual mode persisted throughout the study period (Figure 2E,F). An interannual (2–3 years) mode is also observed in local WPS in the southern area during years 2007–2011; although significant, its power was not as high as for the 1 year mode. Finally, the local WPS for SST revealed the presence of an annual periodic component between 2003–2017 for the northern (SST–North, Figure 2G) and southern areas (SST–North, Figure 2H). The WPS of climatic indices showed different patterns of temporal variability. The SAM showed one main periodic component of 2-year between 2008–2010, while a 3–4 years signal was also present between 2003–2015 (Figure 3A). The local WPS of the monthly time series of SOI presented a unique and significant 2–3 years periodic component from 2006 to 2014 (Figure 3B).

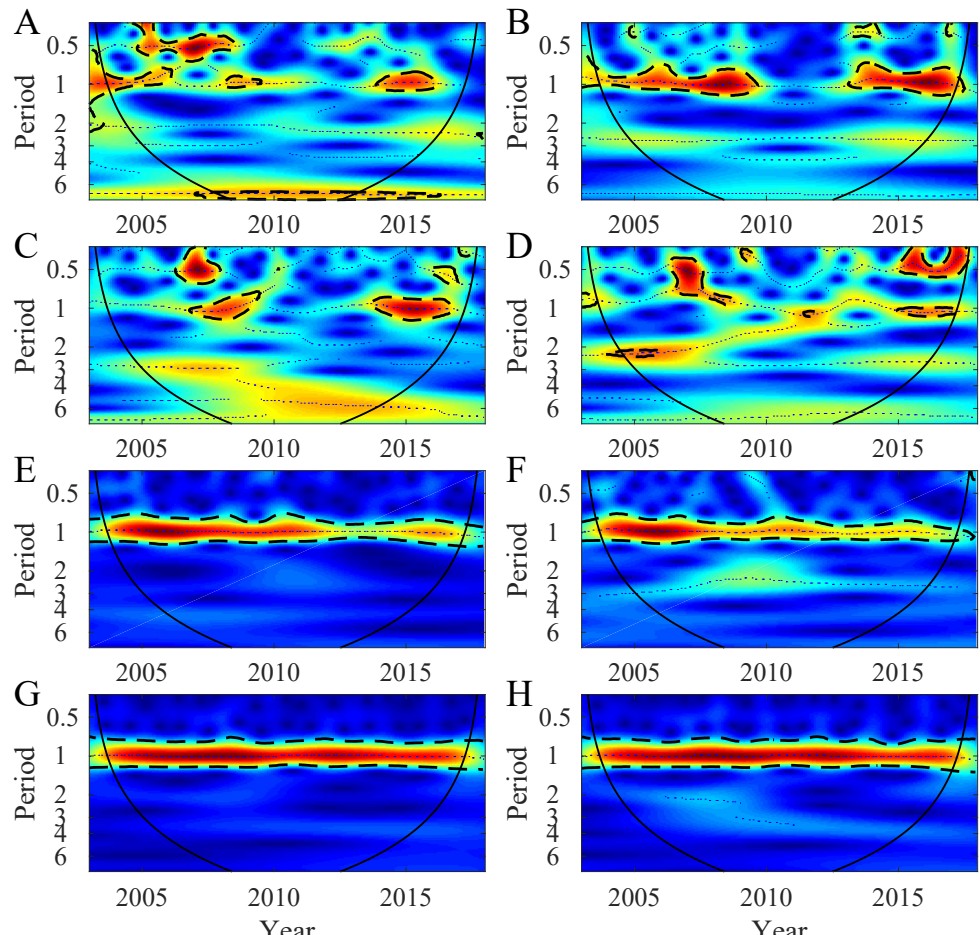

**Figure 2.** Wavelet Power Spectrum (WPS) showing the dominant periods (in years) of variability for the 2003–2017 time series. (**A**) Chl–a in northern area. (**B**) Chl–a in southern area. (**C**) nFLH in northern area. (**D**) nFLH in southern area. (**E**) EVI in northern area. (**F**) EVI in southern area. (**G**) SST in northern area. (**H**) SST in southern area. The black line define the cone of influence above which computations are not influenced by edge effects. The colour code for power values is graded from blue (low power) to red (high power). The black dot–dashed lines indicate the 95% and 90% significant areas obtained by adapted bootstrapping [55].

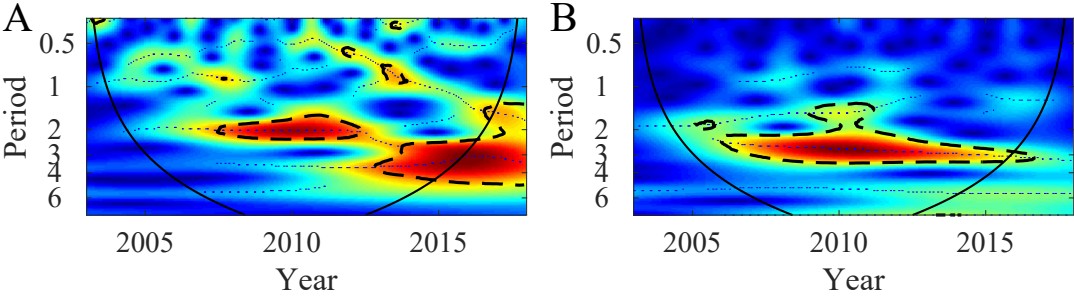

**Figure 3.** Wavelet Power Spectrum as defined in Figure 2 but for (**A**) SAM and (**B**) SOI.

## 3.2. Synchrony in the Northern–Patagonia Ecosystem

The wavelet coherence analyses (supplementary information; S1, S2 and S3) showed that the seasonal mode for all biological variables derived from satellite data (i.e., Chl–a, nFLH and EVI) is shared with local environmental forcing (i.e., SST, S1A, S2A and S3A). Synchrony between the biological variables with climatic-scale variability in both regions showed significant but transient oscillations. Again, these oscillations, particularly over long temporal scales, must be interpreted carefully as the time series used in the analysis are relatively short.

Using PWC, we sequentially controlled firstly, for the influence of climatic-scale variability (i.e., SAM and SOI), on the synchrony between SST and biological variables. Then we controlled for the influence of SST and SOI on the synchrony between SAM and biological variables, and finally, we controlled for the effects of SST and SAM on the effects of SOI on biological variables. Results of the PWC are presented in Figures 4–6. Firstly, Chl-a, showed well-defined coherence with SST over the annual period in both the northern and southern areas (Figure 4A,B). When we examined coherence between Chl–a and SAM, we observed significant coherence in the intra-annual band (0.5-year) throughout the study period in both study areas (Figure 4C,D). Similarly, significant biannual coherence was apparent for both regions after 2008 and over longer (interannual; 4–6 years) periods throughout the records (i.e., 2007 to 2014). Finally, coherence between Chl–a and SOI (Figure 4E,F) was restricted to modes longer than 2–3 years over the period before 2011, suggesting that SAM and SOI exert their influence over different time scales in the northern and southern areas.

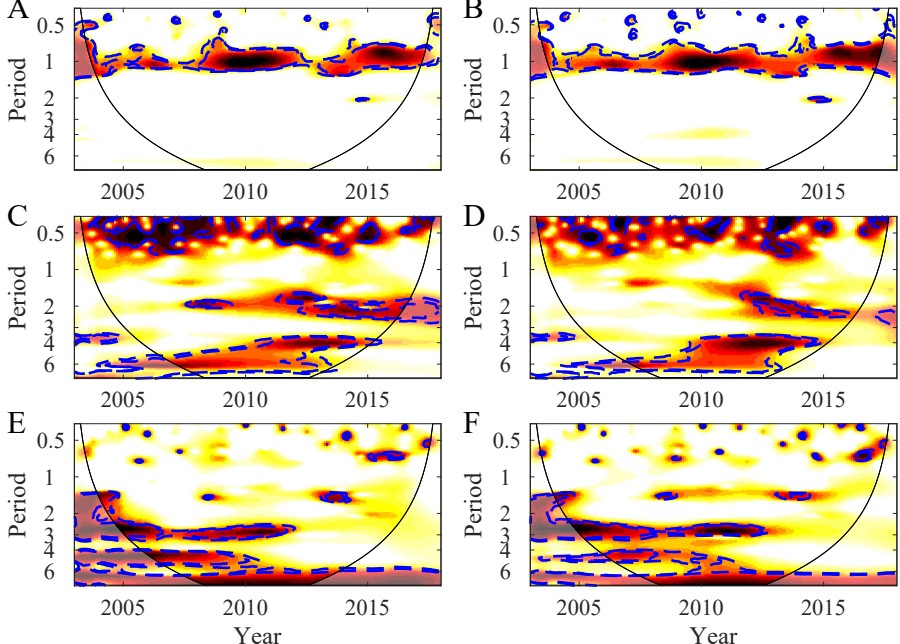

**Figure 4.** Partial Wavelet Coherence (**A**) between Chl–a–North and SST–North controlled by SAM and SOI. (**B**) between Chl–a–South and SST–South controlled by SAM and SOI. (**C**) between Chl–a–North and SAM controlled by SST–North and SOI. (**D**) between Chl–a–South and SAM controlled by SST–south and SOI. (**E**) between Chl–a–North and SOI controlled by SST–North and SAM. (**F**) between Chl–a–South and SOI controlled by SST–South and SAM. Blue dashed lines indicate the 95% and 90% significant areas obtained by adapted bootstrapping [55] and the cone of influence (solid black lines) indicates the regions where the wavelet computations are not influenced by edge effects. The colour code for synchrony values is graded from yellow (low synchrony) to red (high synchrony). Periods of variability (in years) of the $y$-axis are on $\log_2$ scale.

In the case of nFLH–North and SST–North, apparent coherence was observed over the annual band across the study period (Figure 5A), whereas in the southern sector significant coherence was evident from 2008 to 2016 (Figure 5B). PWC between nFLH and SAM revealed significant coherence over the intra-annual band (0.5-year) throughout the study period in the northern and southern sectors in addition to transient coherence over the 2-year mode in the period 2012–2014 in the northern zone and 2007–2015 in the southern zone. Similar interannual coherence was observed for both study regions between 2008–2013 (Figure 5C,D). Significant PWC between nFLH and ENSO was restricted to the interannual to multiannual (from 2 to 6-year) band from 2005–2004 until 2012 (Figure 5E,F).

The PWC between the terrestrial biological signal (EVI) and SST revealed a single relevant scale (1-year mode) throughout the study period in both areas (Figure 6A,B). The PWC analysis between EVI and SAM revealed intra-annual (0.5-year) and interannual (2–4 years) transient modes (Figure 6C,D). Finally, PWC between EVI and SOI showed coherence over the interannual mode over temporal periods similar to nFLH and SOI (Figure 6E,F, periods > 2-year).

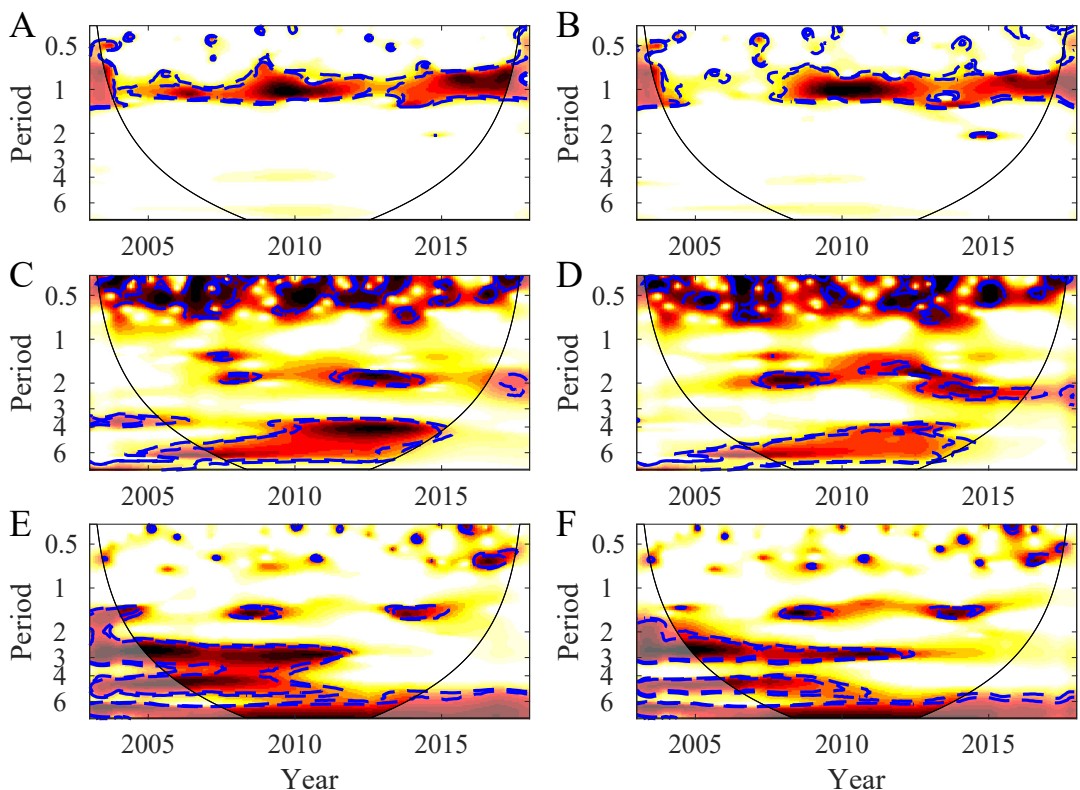

**Figure 5.** Partial Wavelet Coherences as defined in Figure 4 but for nFLH.

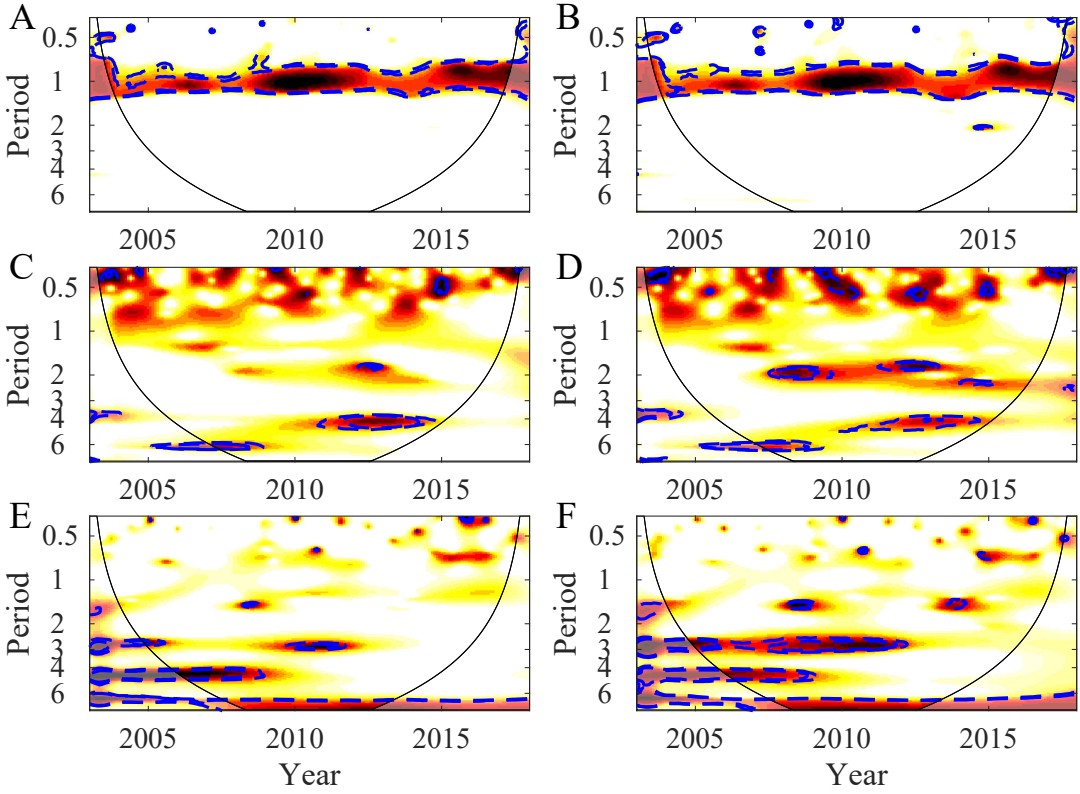

**Figure 6.** Partial Wavelet Coherences as defined in Figure 4 but for EVI.

## 4. Discussion

Our results demonstrated a tight regional association through significant synchrony between biological production and environmental forcing over multiple temporal scales. Moreover, these relationships suggest the presence of subtle differences between the two adjacent regions examined. Below we discuss the potential mechanisms through which climatic and local environmental controls modulate biological responses and the implications of climate-driven changes on primary productivity.

Phenological transitions mark the seasonal progression of autotrophic biomass (i.e., timing), which are modulated by changes in the rates of photosynthesis in both marine and terrestrial ecosystems [56]. Vascular plants in terrestrial ecosystems have a specific period of one year, which is similarly conserved in marine systems, but can experience important local modulation [16,57,58]. Thus, phenological flexibility may generate a productivity mosaic in the timing of the joint biological response to climatic variability observed in our study system [57,59]. The terrestrial ecosystems of Northern Patagonia are dominated by endemic Valdivian and North-Patagonian evergreen forests typical of mid-latitudes [60,61]. The soils of the region have an impervious hard pan (50–60 cm depth), which allows for a year-round water supply following the high levels of precipitation during the winter months [61]. Then, temporal variation in primary productivity follows a pattern similar to the atmospheric temperature [60].

Primary biological production in the adjacent marine ecosystem, (measured as surface Chl–a concentration and nFLH) is dominated by a seasonal mode distinctively associated with local SST control. Variability over higher frequencies and the 2-year mode after 2011 appears to be linked to SAM, whereas more extended periods of oscillation, such as 2–3-year mode before 2011 and the 4–5 years mode appear to be associated mainly to SOI, while after 2011 SAM appears as an essential contributor over the same temporal scales. The higher biological activity in the northern area could be related to

coastal topography and the presence of a cluster of smaller islands driving strong physical-biological coupling in the sector [16,24,62]. Iriarte et al. [63] showed significant associations between autotrophic abundance and streamflow of the Puelo River, which drains in the central part of the northern sector.

A disruption of the annual cycle in surface Chl–a concentration and nFLH took place during the early 2010's across the study region. This breakdown of the seasonal cycle and its relation to the antiphase between SAM (negative phase) and ENSO has been reported for the region before [16,24]. Briefly, after a prolonged near-neutral period during the early 2000's, the SAM experienced positive anomalies, firstly during the observed multidecadal antiphase with ENSO, and culminating with an in-phase period during the 2015–2016 warm ENSO event [39,64]. The latter climate pattern has been related to several phenological anomalies (e.g., anomalous timing of algal blooms), and local environmental disruption in the Inner Sea of Chiloé observed during the summer of 2016 [29,64]. These studies argue that changes in local environmental patterns, forced by anomalies in large-scale modes of atmospheric circulation (i.e. SAM and ENSO), are of major importance for local patterns of primary production and impacting the aquaculture industry, which provides valuables food and goods for human population [65].

The properties of vegetation indices derived from MODIS provide a substantially improved dataset for vegetation monitoring [34,41]. Recently, Lara et al. [39] showed that EVI efficiently captures inter and intra-annual phenology pattern in Chiloé Island ecosystem. In this study, the authors suggested that the weakening in the spectral power of the vegetation activity in both northern and southern sections could be related to the change in the ocean-atmosphere conditions in the Pacific Ocean. The change in phase (from cooling to warming in Ocean temperature) from late-2014 is associated with the Madden-Julian Oscillation (MJO) activity that led to a strong El Niño event during 2015 [66]. The influence of MJO over regional climate in South-America induces changes in surface temperature and rainfall [67], which may have modified the phenological cycle in this highly productive temperate ecosystem dominated by the annual cycle. Prior to the El Niño event of 2015, neutral-ENSO conditions prevailed during the 2011–1014 period, generating a sustained 50–70% deficit in rainfall over the central Chile region dubbed the "Central Chile Megadrought" [68,69]. These studies report a major climatic influence (e.g., Pacific Decadal Oscillation–PDO) on local processes (i.e. rainfall variability, river streamflow), modifying the seasonal variations of EVI as a proxy of gross plant productivity. The climatic variation pattern can modify the matching seasonality between terrestrial and marine ecosystems in northern Patagonia, where the local climate is similarly driven by large-scale circulation [23].

The main question that arises from this study concerns the dynamics of the phenological cycle in a highly productive temperate ecosystem: can we attribute the pattern of temporal variability of EVI and Chl-a (and nFLH) to a response to climatic variability? In the northern Patagonia ecosystem, the seasonality of marine and terrestrial ecosystems was briefly interrupted during the early 2010's period, possibly due to a combination of climate factors. The effect of climatic fluctuations has been previously reported in diverse ecosystems. For instance, changes in synchrony in multiple zooplankton groups and SST was found in the North Sea [11], while on a global scale the synchrony of terrestrial vegetation is determined by temperature and precipitation [14], and the synchrony in the global oceanic phytoplankton biomass is determined by SST [2]. These and other studies at different scales reveal that this process is ubiquitous and operates at multiple scales [8]. We observed that the strength of synchrony varies depending on the frequency band (i.e., period or timescale) with subtle differences between the northern and the southern section of northern Patagonia ecosystem. The frequency-dependence underscores the importance of considering different time scales [2,11,14]. The mechanisms for the different responses in frequency, chiefly persistent synchrony between primary production and SAM in the southern sector post-2010, could be attributed to the different cooling patterns around our study region [62]. The area to the south of Desertores Islands (see Figure 1B) is colder than the northern area due to its direct connection with oceanic SST patterns, especially during the autumn and winter months. Strub et al. [62] reports on strong topographic and hydrographic



mixing around Desertores island, which we hypothesize can act as a spatial barrier to the various synchrony patterns observed in our analysis.

## 5. Conclusions

To our knowledge, this is the first study documenting the synchrony of biological variables coupled with environmental and climatic variability in the ecosystem of northern Patagonia. The biological dynamics in both marine and terrestrial systems present different scales of variability after controlling for the different environmental variables; a clear intraseasonal coupling with SST is observed after controlling for large-scale climatic variability, while at subannual and interannual scales these relationships are driven climatically by ENSO and SAM, which exert a strong influence in northern Patagonia. Our analyses of the temporal patterns between phenological variability and climatic oscillations suggest a complex dynamics between phenology and climatic oscillations in the northern Patagonia ecosystem. Thus, more systematic research is needed to better understand long-term variability in biological production and its correlation with climatic factors.

**Author Contributions:** Conceptualization, C.L., B.C. and B.R.B.; Data curation, B.C. and R.P.F.; Formal analysis, C.L. and B.C.; Funding acquisition, G.S.S. and B.R.B.; Investigation, C.L., G.S.S. and Á.L.P.; Methodology, C.L., B.C., R.P.F. and Á.L.P.; Project administration, C.L.; Resources, G.S.S.; Software, C.L., B.C., R.P.F., Á.L.P. and B.R.B.; Visualization, C.L., B.C., G.S.S. and B.R.B.; Writing—original draft, C.L., B.C., G.S.S., R.P.F., Á.L.P. and B.R.B.; Writing—review & editing, C.L., B.C., G.S.S., R.P.F. and B.R.B.

**Funding:** This study was partially supported by the Millennium Nucleus Center for the Study of Multiple–Drivers on Marine Socio–Ecological Systems (grant ICM MUSELS NC120086) and the Bioengineering Innovation Center, Facultad de Ingeniería y Ciencias, Universidad Adolfo Ibañez. G.S.S. was partially supported by an NSERC Banting Postdoctoral Fellowship. Additional support from grant FONDECYT 1181300 and 1190529 to B.R.B. and 1190805 to G.S.S. is also acknowledged.

**Acknowledgments:** Level–1 MODIS-Aqua files are available from NASA ocean color website (https://oceancolor.gsfc.nasa.gov/). The vegetation indices from the MODIS-TERRA data product were retrieved from the online Data Pool, courtesy of the NASA Land Processes Distributed Active Archive Center (LP DAAC), USGS/Earth Resources Observation and Science (EROS) Center, Sioux Falls, South Dakota. SOI data can be obtained from NOAA website https://www.cpc.ncep.noaa.gov/data/indices/soi whereas SAM time series are available at https://legacy.bas.ac.uk/met/gjma/sam.html.

**Conflicts of Interest:** The authors declare no conflict of interest.

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
