# Peer review of "Coupled Biospheric Synchrony of the Coastal Temperate Ecosystem in Northern Patagonia: A Remote Sensing Analysis"

_remotesensing, doi:10.3390/rs11182092_

Round 1

Reviewer 1 Report

Dear authors,

line 51: You must include the FAO, 2008 in the reference list.

Figure 1.B must be included in the methodology paragraph. and the legend: nflh be changed for capital letters. The Statistical analysis paragraph of Figure 1B must be included in section 2.4.

line 99: The idea must be improved: "River discharge follows a nival regime and comes primarily from Reloncavi and Comau fjords,....".

line 168: The authors could add 2.5 Synchrony analysis by wavelet method subtitle because the synchrony in the northern Patagonia Ecosystem is the core of the paper.

line 188: The normalized SST for the north and south part must show the multi-annual component for the 2010-2013 period because the annual frequency of the SST explain over 90% of the variability of the time series. We strongly recommend using normalized SST. 

line 318-321: the conclusion "clear intraseasonal synchrony with SST is observed", could be changed with analysis the normalized SST. We strongly recommend exploring the temporal patterns on interannual variability of SST.

Author Response

Point 1. line 51: You must include the FAO, 2008 in the reference list.

Response. Done, the reference (FAO, 2018) was included.

Point 2. Figure 1.B must be included in the methodology paragraph. and the legend: nflh be changed for capital letters. The Statistical analysis paragraph of Figure 1B must be included in section 2.4.

Response. Done.

Point 3. line 99: The idea must be improved: "River discharge follows a nival regime and comes primarily from Reloncavi and Comau fjords,....".

Response. Thanks for the observation. The idea was improved as “River discharge follows a nival regime and comes primarily from the Petrohué, Cochamó, Puelo, Riñihue and Yelcho Rivers, located in our study region.The combined freshwater discharge from these rivers influences local ocean circulation and the spring–autumn phytoplankton bloom dynamics (Iriarte et al 2007,León Muñoz et al, 2018).” Please see lines:103--106

Point 4. line 168: The authors could add 2.5 Synchrony analysis by wavelet method subtitle because the synchrony in the northern Patagonia Ecosystem is the core of the paper.

Response. Done. A sub-section 2.5 “Coherence and Synchrony analysis” was added. Please see line 169.

Point 5. line 188: The normalized SST for the north and south part must show the multi-annual component for the 2010-2013 period because the annual frequency of the SST explain over 90% of the variability of the time series. We strongly recommend using normalized SST.

Response. Thanks for the observation. Indeed, all time series were standardized prior to analysis. This is now stated in the legend of Figure 1 and at the end of the Methods section.

Point 6. line 318-321: the conclusion "clear intraseasonal synchrony with SST is observed", could be changed with analysis the normalized SST. We strongly recommend exploring the temporal patterns on interannual variability of SST.

Response. Please see the earlier comment, using standardized time series is a regular procedure for wavelet and synchrony analyses, thanks for highlighting it.

Reviewer 2 Report

Overview and general recommendation

Investigating temporal changes in primary producers is important for understanding ongoing and future effect of environmental changes worldwide. Limited attention has been given to changes in synchrony over time either in marine or terrestrial ecosystems. This study uses 15 years of moderate spatial and temporal resolved data from MODIS (Aqua and Terra) to investigate the relationship between temporal changes in terrestrial and marine primary producers to environmental and climatic variability across northern Patagonia. The study uses Enhanced Vegetation Index (EVI), surface chlorophyll a concentration (Chl–a) and normalized fluorescence line height (nFLH) as indicators of terrestrial and marine primary production. The region was divided in two areas where time series were extracted. Wavelet analysis is used to look at temporal variability and synchrony between the primary producers, SST and climate fluctuations (ENSO and SAM). They concluded that there is a complex dynamics between phenology and climatic oscillations in the northern Patagonia ecosystem.

The study is relevant as there are not many works on this topic in the region. The text is overall clear and easy to read but details are missing and the discussion could be improved. I believe that some points need to be better addressed and explained before publication on RS.

Major comments:

1. The term phenology is misused in the text. Phenology refers to life cycle events, “the timing of seasonally recurring life-history events” as described in Keogan et al. (2018), like flowering time, length of the growing season, or start of the phytoplankton bloom. The manuscript investigates the temporal variability of primary producers, but not their phenological activity. Please revise the use of this term throughout the text (e.g. Line 73).

Keogan et al. 2018. Global phenological insensitivity to shifting ocean temperatures among seabirds. Nature Climate Changevolume 8, pages313–318 (2018).

2. Clarify and emphasize in the text what the authors mean with synchrony. The term is often used in the manuscript but from a first read it looks like a time series study and not really on the synchrony. For example, it is written in the conclusion “the biological dynamics in both marine and terrestrial systems present different scales, a clear intraseasonal synchrony with the SST is observed” Is the study about the synchrony between primary producers or between primary producers and environmental fluctuations? I got confused.

3. The introduction is well written but I missed the word “ENSO”, “SAM”. Although it is easy to guess, the first time it was written which climate fluctuations were investigated was in methods apart from the abstract. Be more specific. For example in line 56-57, “…. which was linked to large-scale climate variability…” which one?

4. Change the figures’ position in the text and put them after they are cited. In latex you can use the “float” package and then the [H] option (e.g. \begin{figure}[H]).

5. Figure 1:

- show the location of the region in a larger map;

- use σ as symbol of standard deviation in the colorbar of figure 1A and y label of Figure 1B;

- all the description on the time series should be in section 2 (“ The full…”). I guess the same methodology was applied to Chl-a data? Add all the description of how the time series were obtained as a separate topic or in 2.3.

- remove the term “in order”, it appears three times and became repetitive;

- clarify in the text or overall in the captions that the time series correspond to standardized anomaly values.

- the histogram (Figure 1B) could be removed if it is not mentioned/discussed in the text.

6. Why only choosing SST as environmental factor?

7. Add more information about the satellite products, mainly ocean colour and SST products. Which algorithms were used? Which flags were applied? What is the temporal and spatial resolution? What nFLH represents? Mind that the nFLH retrieval is impacted by the presence of elevated sediment concentrations.

8. The caption of Figures 5, 6 and 7 are the same except for Chl-a, nFLh and EVI. It is enough to write it once in Fig. 5 and mention in the caption of Figures 6 and 7 something like: “Partial Wavelet Coherence as defined Fig 5 but for nFLH.” The same for the supplementary material and figure 3.

9. It helps the reader if there is an indication which variable the plot refers to: SST, SOI, Chl-a, also if it is north or south. Could be “Chl-a - North”, “SST - South”. The same for the plots in the supplement, e.g. “Chl-a – SST”.

10. If I understood it right, in the wavelet coherence analysis the results can be influenced by the effect of other variables, but using the partial-coherence analysis you can remove the effect of the other variables. What is the reason of showing the wavelet coherence analysis the results? I would suggest removing them.

11. Overall in the discussion, what is the link between ENSO/SAM and the primary producers? Are there studies explaining how different phases of SAM and ENSO influence the region? How these climatic perturbations/SST can influence phytoplankton for instance? There is some discussion on EVI but not for phytoplankton (Chl-a and nFLH).

Minor comments:

Title: remove the point at the end of the title.

Line 51: check the citation format and add FAO 2018 to the references.

Line 114: Color with capital C.

Lines 121-122: I would rather finish the sentence at “increased the noise”. The fact that by using the SWIR bands there are more missing data is not bad, it means that the data was flagged and should not be used.

Lines 122-123: “This is relevant…”. How is this relevant to the present study since the data used was averaged over a large region? Explain it or remove the sentence.

Lines 125-126: Chl-a and nFLH are not proxys of primary production but rather of phytoplankton biomass. Please, revise the text accordantly.

Lines 127-128: Daytime, nighttime?

Lines 179-181: these lines should be in the description (step by step) of how the time series were extracted.

Line 187: change are to were, as the authors used past tense everywhere else in the paragraph.

Lines 188-189: semiannual and annual, right?

Line 190: change a that to that a.

Lines 207-209 and 216-217: there is no need to repeat why wavelet coherence analysis and partial wavelet coherence analysis were used in this study or add the sentences in the methods in a new paragraph after Line 158.

Line 233: and over 4-6 yr periods throughout the records (i.e. 2007 to 2014).

Lines 245-246: I could not identify very distinct patterns of climatic and local environmental forcing between northern and southern areas.

Line 257: remove the space between ( and 50.

Lines 266-269: from what is written (without reading the papers cited) it is hard to understand how these lines are related to the results presented in the manuscript. Here could be discussed how SST, SAM and ENSO can influence phytoplankton in the region.

Lines 271-272: I think that the disruptions in the annual cycle is also evident in nFLH, especially in Fig 2C.

Lines 272-274: here it could be explained how these disruptions are related to SAM and ENSO based as well on your results. How do you know that it is a negative SAM phase and positive ENSO phase. Did you identify a specific strong event? It would be interesting to have a time series plot of the climate indices.

Line 275: what does ISC stand for?

Line 286: change this to the, or specify “change in phase” of what? Ocean temperature?

Line 293: what does PDO stand for?

Line 307: which process?

Line 308: which phenomenon? Is the process mentioned before? Keep the same term.

Lines 308-310: I do not see much difference between northern and southern areas in the plots.

Line 311: check citation format.     

Lines 311-312: where are the Desertores Islands?

Lines 313-315: is there a reference or is it an assumption from the present study?

Line 322: why only SAM? Figures 4, 5 and 6 show also very strong power in the plots with SOI.

Author Response

Overview and general recommendation

Investigating temporal changes in primary producers is important for understanding ongoing and future effect of environmental changes worldwide. Limited attention has been given to changes in synchrony over time either in marine or terrestrial ecosystems. This study uses 15 years of moderate spatial and temporal resolved data from MODIS (Aqua and Terra) to investigate the relationship between temporal changes in terrestrial and marine primary producers to environmental and climatic variability across northern Patagonia. The study uses Enhanced Vegetation Index (EVI), surface chlorophyll a concentration (Chl–a) and normalized fluorescence line height (nFLH) as indicators of terrestrial and marine primary production. The region was divided in two areas where time series were extracted. Wavelet analysis is used to look at temporal variability and synchrony between the primary producers, SST and climate fluctuations (ENSO and SAM). They concluded that there is a complex dynamics between phenology and climatic oscillations in the northern Patagonia ecosystem. The study is relevant as there are not many works on this topic in the region. The text is overall clear and easy to read but details are missing and the discussion could be improved. I believe that some points need to be better addressed and explained before publication on RS.

Response. Thank you for your recognition to our contribution.

Major comments:

Point 1. The term phenology is misused in the text. Phenology refers to life cycle events, “the timing of seasonally recurring life-history events” as described in Keogan et al. (2018), like flowering time, length of the growing season, or start of the phytoplankton bloom. The manuscript investigates the temporal variability of primary producers, but not their phenological activity. Please revise the use of this term throughout the text (e.g. Line 73).

Keogan et al. 2018. Global phenological insensitivity to shifting ocean temperatures among seabirds. Nature Climate Change volume 8, pages 313–318 (2018).

Response. We agree with the reviewer that our use of the term is confusing since we intend to show the cyclical dynamics of biological activity, i.e. the phenological cycle. We have edited the manuscript to emphasize that the study is focused on the study of continuous cyclical processes.

Point 2. Clarify and emphasize in the text what the authors mean with synchrony. The term is often used in the manuscript but from a first read it looks like a time series study and not really on the synchrony. For example, it is written in the conclusion “the biological dynamics in both marine and terrestrial systems present variability different scales, a clear intraseasonal synchrony with the SST is observed” Is the study about the synchrony between primary producers or between primary producers and environmental fluctuations? I got confused.

Response. We acknowledge the sharp eye of the reviewer, we have edited the manuscript to assure that synchrony refers to the coherent time-dependent variation between physical and biological variables, and also rephrased our explanation in subsection: Synchrony in the northern--Patagonia Ecosystem.

Point 3. The introduction is well written but I missed the word “ENSO”, “SAM”. Although it is easy to guess, the first time it was written which climate fluctuations were investigated was in methods apart from the abstract. Be more specific. For example in line 56-57, “…. which was linked to large-scale climate variability…” which one?

Response. We appreciate the observation. We now explicitly include the relationship between ENSO, SAM and PDO climate indices from previous studies about the causes of temporal variability of chlorophyll-a concentration in the study area. Please see lines 50-62.

Point 4. Change the figures’ position in the text and put them after they are cited. In latex you can use the “float” package and then the [H] option (e.g. \begin{figure}[H]).

Response. Done. Thanks.

Point 5. Figure 1: show the location of the region in a larger map; use σ as symbol of standard deviation in the colorbar of figure 1A and y label of Figure 1B;

Response. Done.

Point 6 all the description on the time series should be in section 2 (“ The full…”). I guess the same methodology was applied to Chl-a data? Add all the description of how the time series were obtained as a separate topic or in 2.3.

Response. We added the information in subsection 2.4.

Point 7 remove the term “in order”, it appears three times and became repetitive;

Response. We edited the whole section to improve clarity.

Point 8 clarify in the text or overall in the captions that the time. series correspond to standardized anomaly values.

Response. Done.

Point 9 the histogram (Figure 1B) could be removed if it is not mentioned/discussed in the text.

Response. We have added a mention to the histogram at the end of the methods section.

Point 10. Why only choosing SST as environmental factor?

Response. We highlight in the description of the study area, and support with several citations, that SST is a key driver of local weather in the region. Moreover, climatic variability is usually linked to local weather patterns through SST (i.e. ENSO).

Point 11. Add more information about the satellite products, mainly ocean colour and SST products. Which algorithms were used? Which flags were applied? What is the temporal and spatial resolution? What nFLH represents? Mind that the nFLH retrieval is impacted by the presence of elevated sediment concentrations.

Response: Thanks for the observation. We added a paragraph between lines 126-134.

Point 12. The caption of Figures 5, 6 and 7 are the same except for Chl-a, nFLh and EVI. It is enough to write it once in Fig. 5 and mention in the caption of Figures 6 and 7 something like: “Partial Wavelet Coherence as defined Fig 5 but for nFLH.” The same for the supplementary material and figure 3.

Response. Done.

Point 13. It helps the reader if there is an indication which variable the plot refers to: SST, SOI, Chl-a, also if it is north or south. Could be “Chl-a - North”, “SST - South”. The same for the plots in the supplement, e.g. “Chl-a – SST”.

Response. Thank you for observations. We have edited the legends and text to follow the figures in order of appearance in the text.

Point 14. If I understood it right, in the wavelet coherence analysis the results can be influenced by the effect of other variables, but using the partial-coherence analysis you can remove the effect of the other variables. What is the reason of showing the wavelet coherence analysis the results? I would suggest removing them.

Response. As this is one of the first papers that use partial wavelet coherence, we wished to put in the Appendix the results obtained with the classical wavelet coherence approach. This also allows the reader to compare the results obtained with the classical approach and those attained with the new one. Nevertheless this is very standard, using partial correlations or partial Fourier coherences.

Point 15. Overall in the discussion, what is the link between ENSO/SAM and the primary producers? Are there studies explaining how different phases of SAM and ENSO influence the region? How these climatic perturbations/SST can influence phytoplankton for instance? There is some discussion on EVI but not for phytoplankton (Chl-a and nFLH).

Response. We have extensively edited the relevant sections in the discussion and made the connections more clear, thanks for constructive observation.

Minor comments:

Title: remove the point at the end of the title.

Response. Done.

Line 51: check the citation format and add FAO 2018 to the references.

Response. Done.

Line 114: Color with capital C.

Response. Done.

Lines 121-122: I would rather finish the sentence at “increased the noise”. The fact that by using the SWIR bands there are more missing data is not bad, it means that the data was flagged and should not be used.

Response. We finish the sentence “… in SeaDas increased the noise”. Please see line 126.

Lines 122-123: “This is relevant…”. How is this relevant to the present study since the data used was averaged over a large region? Explain it or remove the sentence.

Response. Done. These sentences was removed.

Lines 125-126: Chl-a and nFLH are not proxys of primary production but rather of phytoplankton biomass. Please, revise the text accordantly.

Response. Done.

Lines 127-128: Daytime, nighttime?

Response. Thanks for the question. We used daytime SST in order to have concomitant satellite data with nFLH and Chl-a. Please see line 133.

Lines 179-181: these lines should be in the description (step by step) of how the time series were extracted.

Response. Thanks for the observation, we have added a description at the end of the Methods section.

Line 187: change are to were, as the authors used past tense everywhere else in the paragraph.

Response. Done.

Lines 188-189: semiannual and annual, right?

Response. We refer to interannual components. Please see line 204.

Line 190: change a that to that a.

Response. Done.

Lines 207-209 and 216-217: there is no need to repeat why wavelet coherence analysis and partial wavelet coherence analysis were used in this study or add the sentences in the methods in a new paragraph after Line 158.

Response. Done.

Line 233: and over 4-6 yr periods throughout the records (i.e. 2007 to 2014).

Response. Correct, we added it. Please see line 249.

Lines 245-246: I could not identify very distinct patterns of climatic and local environmental forcing between northern and southern areas.

Response. Thanks for the comment. We have toned down our assertion and explained in more detail the differences. Please see lines 261-265.

Line 257: remove the space between ( and 50.

Response. Done.

Lines 266-269: from what is written (without reading the papers cited) it is hard to understand how these lines are related to the results presented in the manuscript. Here could be discussed how SST, SAM and ENSO can influence phytoplankton in the region.

Response. Thank you, we have added a few lines (289-294) regarding prior evidence.

Lines 271-272: I think that the disruptions in the annual cycle is also evident in nFLH, especially in Fig 2C.

Response. Thank you for the observation. We added: “We observed a disruption of the annual cycle in surface Chl--a concentration and nFLH across the study region”. Please see line 289-290.

Lines 272-274: here it could be explained how these disruptions are related to SAM and ENSO based as well on your results. How do you know that it is a negative SAM phase and positive ENSO phase. Did you identify a specific strong event? It would be interesting to have a time series plot of the climate indices.

Response. Thanks for the observation, we have expanded the discussion as suggested in lines 288-290. The time series are presented in Fig. 2, Lara et al. 2018, and in Fig. 2, Garreaud 2018, both of which are referenced in the paragraph.

Line 275: what does ISC stand for?

Response. We changed ISC for Inner Sea of Chiloé. Please see line 296.

Line 286: change this to the, or specify “change in phase” of what? Ocean temperature?

Response. We now specify that these are changes in phase of the ocean temperature “The change in phase (from cooling to warming in Ocean temperature)”. Please see line 307.

Line 293: what does PDO stand for?

Response. We now specify“….(e.g. Pacific Decadal Oscillation–PDO)”. Please see line 314.

Line 307: which process?

Response. Synchrony”.

Line 308: which phenomenon? Is the process mentioned before? Keep the same term.

Response. Please see response to major comment #2

24. Lines 308-310: I do not see much difference between northern and southern areas in the plots.

Response. We have toned down the references for regarding north-south differences and discussed a specific aspect related to a potential driving mechanism. Please see lines 330-332.

25.  Line 311: check citation format.

Response. Done.

26. Lines 311-312: where are the Desertores Islands?

Response. Please see legend in Figure 1.

27.

Lines 313-315: is there a reference or is it an assumption from the present study?

Response. Is a reference and assumption from our study. We changed this sentence as “Strub et al. (2019) report strong topographic and hydrographic mixing around Desertores island, which we hypothesize to be a spatial barrier contributing to the heterogenous synchrony observed in our analysis”. Please see lines 337-339.

28. Line 322: why only SAM? Figures 4, 5 and 6 show also very strong power in the plots with SOI.

Response. We modified this sentence to reflect this result, thanks for the observation.

Reviewer 3 Report

This was an interesting paper using wavelet analysis to find periodicity in various satellite-derived time series and correlations between these time series using wavelet coherence.  The paper was well-written overall and was significant in its conclusions. 

A table of abbreviations for the various time series would be helpful. 

Also, in Figure 1, sigma should be used for the axis label instead of square root of sigma^2. 

There are some other small errors, e.g., "frequence" on line 158 should read "frequency" and there is an extra left bracket in the denominator of the formula for wavelet coherence between lines 161 and 162. 

Otherwise, I feel that the paper is suitable for publication. 

Author Response

Comments and Suggestions for Authors

Point 1. This was an interesting paper using wavelet analysis to find periodicity in various satellite-derived time series and correlations between these time series using wavelet coherence. The paper was well-written overall and was significant in its conclusions.

Response. We appreciate your comments and evaluation of our manuscript. Thanks.

Point 2. A table of abbreviations for the various time series would be helpful.

Response. Done. Please see Supplementary material.

Point 3. Also, in Figure 1, sigma should be used for the axis label instead of square root of sigma^2.

Response. Done

Point 4. There are some other small errors, e.g., "frequence" on line 158 should read "frequency" and,

Response. Thank you for this observation. All small errors were resolved.

Point 5. there is an extra left bracket in the denominator of the formula for wavelet coherence between lines 161 and 162.

Response. Thank you for these observations. It has been corrected.

Point 6. Otherwise, I feel that the paper is suitable for publication.

Response. We appreciate your comments.

Round 2

Reviewer 2 Report

The authors have revised the manuscript accordingly and the new version is clearly improved. I suggest the publication of the manuscript, barring few typo errors that should be corrected before its final publication.

------

Line 211: check space between "South,Figure"

Line 216: same for  "South)showed"

Line 220: and "North,Figure"